# DeepSATA: A Deep Learning-Based Sequence Analyzer Incorporating the Transcription Factor Binding Affinity to Dissect the Effects of Non-Coding Genetic Variants

**DOI:** 10.3390/ijms241512023

**Published:** 2023-07-27

**Authors:** Wenlong Ma, Yang Fu, Yongzhou Bao, Zhen Wang, Bowen Lei, Weigang Zheng, Chao Wang, Yuwen Liu

**Affiliations:** 1Shenzhen Branch, Guangdong Laboratory for Lingnan Modern Agriculture, Key Laboratory of Livestock and Poultry Multi-Omics of MARA, Agricultural Genomics Institute at Shenzhen, Chinese Academy of Agricultural Sciences, Shenzhen 518124, China; mawenlong@caas.cn (W.M.); yfu1116@163.com (Y.F.); rzbyz@163.com (Y.B.); 17596551135@163.com (Z.W.); leibowen@webmail.hzau.edu.cn (B.L.); zhengweigang@caas.cn (W.Z.); wangchao2020@webmail.hzau.edu.cn (C.W.); 2Innovation Group of Pig Genome Design and Breeding, Research Centre for Animal Genome, Agricultural Genomics Institute at Shenzhen, Chinese Academy of Agricultural Sciences, Shenzhen 518124, China; 3School of Life Sciences, Henan University, Kaifeng 475004, China; 4Key Laboratory of Agricultural Animal Genetics, Breeding and Reproduction of Ministry of Education & Key Lab of Swine Genetics and Breeding of Ministry of Agriculture and Rural Affairs, Huazhong Agricultural University, Wuhan 430070, China; 5Kunpeng Institute of Modern Agriculture at Foshan, Chinese Academy of Agricultural Sciences, Foshan 528226, China

**Keywords:** non-coding variants, deep learning, transcription factor binding affinity, cross-species prediction, chromatin accessibility, genomic prediction

## Abstract

Utilizing large-scale epigenomics data, deep learning tools can predict the regulatory activity of genomic sequences, annotate non-coding genetic variants, and uncover mechanisms behind complex traits. However, these tools primarily rely on human or mouse data for training, limiting their performance when applied to other species. Furthermore, the limited exploration of many species, particularly in the case of livestock, has led to a scarcity of comprehensive and high-quality epigenetic data, posing challenges in developing reliable deep learning models for decoding their non-coding genomes. The cross-species prediction of the regulatory genome can be achieved by leveraging publicly available data from extensively studied organisms and making use of the conserved DNA binding preferences of transcription factors within the same tissue. In this study, we introduced DeepSATA, a novel deep learning-based sequence analyzer that incorporates the transcription factor binding affinity for the cross-species prediction of chromatin accessibility. By applying DeepSATA to analyze the genomes of pigs, chickens, cattle, humans, and mice, we demonstrated its ability to improve the prediction accuracy of chromatin accessibility and achieve reliable cross-species predictions in animals. Additionally, we showcased its effectiveness in analyzing pig genetic variants associated with economic traits and in increasing the accuracy of genomic predictions. Overall, our study presents a valuable tool to explore the epigenomic landscape of various species and pinpoint regulatory deoxyribonucleic acid (DNA) variants associated with complex traits.

## 1. Introduction

Genome-wide association studies (GWAS) have been proven to be powerful tools for identifying genetic variants associated with diseases or complex traits in various species [1]. However, a large proportion of genetic variants identified through GWAS are located in non-coding regions [2], which have long been considered “junk DNA” with little functional significance. Recent studies have revealed that non-coding variants can play a critical role in the development of a wide range of human disorders [3,4] or important economic traits in animals [5,6] by dysregulating gene expression. Therefore, the functional interpretation and prioritization of non-coding variants are essential for improving our understanding of the genetic basis of human diseases and complex traits in animals.

The development of high-throughput sequencing technologies has greatly boosted the understanding of functional effects of non-coding sequences. Technologies such as ChIP-seq [7], ATAC-seq [8], and DNase-seq [9] have significant capacities in conducting comprehensive epigenetic investigations at the genome-wide scale. They provide valuable information on how non-coding elements such as enhancers, insulators, and promoters are regulated through transcription factor binding and chromatin accessibility. Currently, considerable epigenome data for diverse species, tissues, and cell types have been generated and are curated in many public databases such as ENCODE [10], Roadmap Epigenomics [11], and FAANG (Functional Annotation of Animal Genomes) [12], enabling the systematic modeling of the intricate mechanisms involved in epigenetic regulation.

However, modeling genomic data presents significant challenges due to the extensive scale and intricate nature. Analytical methods should consider direct or indirect relationships among high-dimensional data and generate appropriate hypotheses to make accurate predictions. Deep learning (DL), a state-of-the-art approach that employs multi-layer artificial neural networks, has been proven effective in deciphering intricate features of multiple types of large-scale genomic data [13,14]. DL has demonstrated its capability in predicting the epigenetic status of deoxyribonucleic acid (DNA) sequences, such as the impact of non-coding variants on splicing, histone marks, and transcription factor (TF) binding [15,16]. Deep neural networks (DNN) are the most prevalent DL algorithms, which have been widely employed in many powerful tools for annotating non-coding variants, such as DeepSEA [16], Basset [17], DeepMILO [18], and DeepFun [19]. Although these tools have proven valuable in exploring regulatory mechanisms, their training predominantly relies on data from humans or mice, rendering them unable or exhibiting poor performance when applied to other species. The genomes of many other species have not been as thoroughly studied and annotated as those of humans and mice. Their epigenetic data are limited and of lower quality, posing a significant challenge for accurate function predictions. The use of cross-species prediction by leveraging the high-quality data from other species is a promising method to tackle this problem.

Although *cis*-regulatory elements, especially enhancers, evolve rapidly and exhibit significant variation across species, the TF binding preferences are under significant constraint within the same tissue [20,21,22]. This is a result of evolutionary pressure to maintain proper gene regulation. If the binding affinity of a TF was to change significantly, it would have a drastic effect on the organism because it would alter the regulation of a large number of genes. This could lead to disruptions in essential biological processes and potentially have detrimental effects on the organism’s development, survival, and reproduction [23]. Previous studies have demonstrated the potential of using DL in conjunction with TF binding for cross-species regulatory effect prediction [20,21]. These studies suggest that when regulatory rules are conserved across species, models trained on data from one species can potentially estimate the regulatory effects of DNA sequences in other species. While the accuracy of cross-species predictions may not be on par with predictions within the same species, they may still provide valuable non-coding DNA annotations when the desired data are unavailable or compromised.

Therefore, in this work, we introduced DeepSATA, a novel DL-based sequence analyzer that incorporates the TF binding affinity for cross-species prediction. This model is built upon the well-established DeepSEA [16], a widely recognized DL-based sequence model that is known for its simplicity and effectiveness in predicting the functional effects of genetic variants. We trained a three-layer DNN to learn the regulatory principles that govern TF binding and chromatin accessibility. The model was trained using multi-tissue ATAC-seq data obtained from pigs, chickens, cattle, humans, and mice, in addition to non-redundant TF binding sites specific to vertebrates. We conducted a performance comparison among DeepSATA, DeepSEA and Basset using a receiver operating characteristic curve (ROC) analysis and calculated the area under the curve (AUC). The results unequivocally demonstrated that DeepSATA outperformed DeepSEA and Basset in terms of accuracy. We also conducted a benchmark study to evaluate the performance of cross-species prediction compared to same-species prediction within the same tissue context. Despite the lower accuracy observed in cross-species predictions compared to same-species predictions, their values and utility are still evident. The application of DeepSATA to predict the effects of pig genetic variants demonstrated a strong correspondence with the GWAS findings. The identification of significant chromatin-affecting variants holds great value in improving genomic prediction accuracy and provides valuable insights into the economic traits of pigs. In summary, our model is able to effectively predict the regulatory function of non-coding genomic regions that are currently poorly understood. This can significantly broaden our understanding of the potential impacts of non-coding variants on complex traits across a wide range of species.

## 2. Results

### 2.1. Overview of DeepSATA

DeepSATA is a user-friendly framework designed for the identification of open chromatin regions (OCRs) throughout the entire genome and the functional annotation of non-coding genetic variants. For a DNA sequence, DeepSATA extends the traditional two-dimensional space of one-hot encoding to a three-dimensional space that incorporates binding configurations of TFs that are important in the specific biological context being studied. These TFs are selected based on the enrichment of their DNA binding motifs within the OCRs. Then, along the new dimension of the model input matrix, each successive two-dimensional layer encodes the binding affinity and location of a specific TF. The encoding strategy is to use the position weight probability matrix of the TF to populate the matrix at its predicted binding site (Figure 1). Due to the explicit modeling of TF binding preference, DeepSATA more effectively learns *cis*-regulatory grammar from extensive ATAC-seq data. Subsequently, by conducting in silico mutation scanning base by base, one can assess the impact of each base substitution on chromatin accessibility.

Specifically, the genome was divided into bins of 200 base pairs (bp). Each bin was assigned a label of 1 if more than half of the 200 bp fell within the ATAC peak regions and a label of 0 if not (Figure 1). Only the open chromatin bins (OCBs), which are defined as being located within at least one peak region, were used for the subsequent analysis. Every labeled OCB along with 400 bp flanking sequences at its two sides was fed into DeepSATA, which consisted of three sequential convolution layers (320, 480, and 960 kernels). Each OCB is represented by a 1000 × 4 × (*N* + 1) three-dimensional matrix, where 1000 represents the sequence centered around the 200 bp bin, 4 represents the four DNA nucleotides (A, G, C, or T), 1 represents the one-hot encoding of the sequence, and N represents the number of TFs that are predicted to bind to the sequence, based on the TF DNA binding motif. To determine N in the training set, we collected non-redundant TF binding sites from JASPAR 2022 database [24] and used FIMO (version 5.4.1) [25] to identify motif presence patterns with the default *p*-value. Then, we ranked the TFs based on their corresponding motif presence over all input bins and selected the top N TFs for model training. In this study, due to computational resource constraints, we set N = 10, ensuring a balanced compromise between accuracy and computational efficiency. In the initial convolutional neural network (CNN) layer, we utilized 320 kernels with a ReLU activation function. To mitigate overfitting, we adjusted the weights to 0.000001 if they were lower than 0 and left them unchanged otherwise. This was followed by a max pooling layer with a size of 4 × (N + 1) to combine the features extracted from various OCB types, including both the one-hot-encoded sequence information and TF binding preference. Lastly, a dropout operation with a rate of 0.2 was applied. For the subsequent two CNN layers, we used 480 and 960 kernels, respectively. The parameters of the ReLU activation function were set the same as in the first layer. The max pooling layers had strides of 4 × 1, allowing for the downsampling of the feature maps. Dropout rates of 0.2 and 0.5 were applied in these layers to further prevent overfitting. The final fully connected layer was utilized to combine the information obtained from the CNN layers, followed by a sigmoid output layer that computed the probability output for each chromatin feature. To increase the versatility of DeepSATA, it supports multiple input formats such as VCF, BED, and FASTA sequences. The output is a vector of probabilities that represents the likelihood of a sequence being an OCR in each chromatin feature. In the end, we assessed the performance of DeepSATA using the ROC and AUC.

### 2.2. DeepSATA Consistently Outperformed DeepSEA and Basset in AUC Results

In our study, we utilized DeepSATA to evaluate its predictive capabilities for chromatin features in five distinct species: mice, pigs, cattle, humans, and chickens. We collected open chromatin accessibility datasets and corresponding transcription factor binding motifs, which are summarized in Table 1 and Appendix A. The performance of the DeepSATA, DeepSEA, and Basset models was assessed for each species by comparing their average AUC values across different chromatin features. DeepSATA demonstrated superior performance across all species, with average AUC values of 0.854, 0.779, 0.772, 0.759, and 0.744 for mice, pigs, cattle, humans, and chickens, respectively (Figure 2A and Appendix A). In comparison, DeepSEA had average AUC values of 0.796, 0.775, 0.769, 0.755, and 0.736, respectively (Figure 2B and Appendix A), while Basset had average AUC values of 0.778, 0.719, 0.768, 0.717, and 0.722, respectively (Figure 2C and Appendix A). It is worth noting that DeepSATA consistently outperformed DeepSEA and Basset in all chromatin features for pigs and mice (Appendix A). The relative improvement was particularly significant in the cerebrum tissue of the mice, where DeepSATA achieved an AUC of 0.829 compared to DeepSEA’s 0.659 for female mice and an AUC of 0.828 compared to DeepSEA’s 0.653 for male mice, representing a relative improvement of over 25%. Interestingly, our findings indicate that the DeepSATA, DeepSEA, and Basset models performed better for the Duroc pig breed compared to other pig breeds (Appendix A). This suggests a superior ability to recognize regulatory functional patterns in the non-coding genomic regions specific to *Duroc pigs*. In the case of cattle, the chromatin feature of the hypothalamus was most effectively captured, with AUC values of 0.887 for DeepSATA, 0.883 for DeepSEA, and 0.895 for Basset (Appendix A). What particularly encouraged us was the remarkable performance of DeepSATA in accurately recognizing the regulatory patterns of the cerebellum in chickens, achieving AUC values as high as 0.972, while DeepSEA achieved 0.971 and Basset achieved 0.953 (Appendix A). Additionally, in the following analysis, we selected DeepSEA as the baseline comparison, since it achieved better prediction performance compared with Basset. Overall, these results clearly demonstrate the effectiveness of DeepSATA in accurately identifying the regulatory patterns of chromatin features in different animal species.

### 2.3. DeepSATA Achieved Reliable Cross-Species Predictions

The convincing performance of DeepSATA encouraged us to conduct a benchmark study to evaluate its performance of cross-species prediction in comparison to same-species prediction within the same tissue context in animals. To accomplish this, we utilized the DeepSATA model trained on mice, which showed the highest average performance (AUC = 0.854) in same-species predictions (Figure 2 and Appendix A). We utilized this model for cross-species predictions in pigs, chickens, and cattle. In a similar manner, we utilized the corresponding DeepSEA model trained on mice for cross-species predictions on the same set of animals to establish a baseline for comparison. Since the datasets for pigs, cattle, and chickens consisted of male individuals, we utilized the model trained on male mice for cross-species predictions. The overall results demonstrated that DeepSATA outperformed DeepSEA in this scenario (Figure 3). This superiority in performance aligned with the observed trend in same-species predictions (Figure 2 and Figure 3 and Appendix A). Both DeepSEA and DeepSATA showed lower, yet reliable, AUC values compared to same-species predictions (Figure 3 and Appendix A). Moreover, for several specific chromatin features such as the liver in Enshi (ES) pigs, large white (LW) pigs, cattle, and chickens, as well as spleen and muscle tissues in chickens, the performance of cross-species predictions using the DeepSATA model was even better than that of same-species predictions (Figure 3). The relative improvement could be as high as 12.2% (0.725 vs. 0.646) for the liver in chickens (Figure 3 and Appendix A), indicating the flexible and efficient usage of the DeepSATA model across different species.

### 2.4. DeepSATA Achieved Significant Accuracy for Biological Functions

Similar to DeepSEA, DeepSATA possesses the ability to prioritize functional single-nucleotide polymorphisms (SNPs) based on their predicted chromatin effect. We conducted a comprehensive analysis of approximately 11.6 million genome-wide SNPs in around 3000 pigs, utilizing both DeepSATA and DeepSEA. Given the critical significance of skeletal muscle traits in pigs, we directed our investigation towards the influence of SNPs on open chromatin in skeletal muscle tissues. Among the various skeletal muscle models, the Duroc skeletal muscle demonstrated the highest performance, making it an excellent candidate for further examination (refer to Figure 4A,B). Moving forward, we assessed the impact of each SNP base substitution on the accessibility of Duroc skeletal muscle chromatin. To distinguish between open chromatin and non-open chromatin, we employed threshold values (0.256 for DeepSATA and 0.239 for DeepSEA) that yielded an optimal balance between sensitivity and specificity measures (Figure 4C,D). SNPs with alleles displaying distinguishable chromatin states were identified as statistically significant. The DeepSATA successfully identified 70,279 SNPs whose allele alternations had a significant impact on the open chromatin accessibility, whereas DeepSEA identified 75,016 SNPs. Approximately 9.6% (6756) of the clusters identified by DeepSATA were found to be shared with DeepSEA. Moreover, 98.0% of the SNPs (68,884) from the DeepSATA cluster and 98.1% (73,604) from the DeepSEA cluster overlapped within the 2 Mb genomic regions centered at the midpoints of trait-associated pig GWAS signals. To associate these trait-associated SNPs with genes, we assigned them to their closest genes and discovered that 81.5% (9323) of the DeepSATA genes were shared with DeepSEA. Genes specifically linked to trait-associated SNPs in the DeepSATA cluster showed significant enrichment in four traits, with the “texture” trait (false discovery rate (FDR) = 0.001, fold of enrichment = 1.19) being the most significantly enriched (Figure 4E and Appendix A). This trait is an important pork quality measurement and is closely associated with muscle fiber features [26,27]. On the other hand, genes specifically associated with DeepSEA SNPs showed significant enrichment for three traits (Figure 4E and Appendix A). The most significant one was “meat color” (FDR = 0.025, fold or enrichment = 1.16), which is directly influenced by the characteristics of the muscle tissue [28]. The “texture” trait was also significantly enriched but with less significance (FDR = 0.032, fold of enrichment = 1.13) compared to that of DeepSATA (Figure 4E and Appendix A). Overall, DeepSATA, similar to DeepSEA, has the ability to achieve substantial accuracy for biological functions.

### 2.5. Case Study of a Regulatory SNP Identified by DeepSATA

In pork production, back fat is a significant trait, and understanding the genetic mechanisms behind it is crucial in pig research. A recent GWAS study identified a causal regulatory variant (chr5:66103958: G > A [*susScr11*]) within an intron of the *CCND2* gene, influencing its expression [29]. *CCND2* is a gene associated with the cell cycle and has been identified as a strong candidate gene in the deposition of back fat in pigs [29]. Elevated expression of *CCND2* has a suppressive effect on adipogenesis [29]. The G allele has been associated with increased deposition of back fat, while the A allele has been linked to a decrease in back fat [29]. In our study, we investigated the impact of this SNP using the DeepSATA model trained on pigs. The results indicated that both the G and A alleles in fat samples from four pig breeds surpassed the threshold of the optimal balance between sensitivity and specificity (Figure 5A), indicating the presence of open chromatin. Additionally, the A allele consistently had higher scores compared to the G allele across different pig breeds (Figure 5B). This indicates that the A allele has a stronger ability to promote open chromatin. These superior effects on open chromatin suggest the presence of a potent enhancer element, which may subsequently lead to an increased expression of the *CCND2* gene, thereby resulting in the suppression of adipogenesis.

### 2.6. DeepSATA-Improved Phenotype Prediction Performance

After analyzing the potential causal regulatory variant underlying the back fat trait in pigs, we were intrigued by the possibility of leveraging DeepSATA-based regulatory variants to increase the genomic prediction accuracy. Along this direction, we employed the widely used rrBLUP method for genomic prediction but selected different sets of SNPs as inputs for the prediction model [30]. We aimed to predict the back fat thickness at 100 kg, an important economic trait in pig breeding. We used the phenotype data for 1940 *Duroc pigs* from a previous study [31]. The genotype data, which included over 11 million SNPs for each pig, were obtained from the same source [31]. Three sets of input genotypes were used for phenotype prediction: array SNPs (32,451), DeepSATA-based regulatory SNPs (70,279), and DeepSEA-based regulatory SNPs (75,016) (see Section 4 for details). There was a limited overlap observed among these sets, as depicted in Figure 6A. Specifically, a mere 0.019% (32) of SNPs were found to appear in all three sets, while 4.96% (6765) were identified by both DeepSATA and DeepSEA. This indicates that the pools of SNPs for phenotype prediction are nearly non-redundant. Through 5-fold cross-validation, we found that both the DeepSATA (average Pearson correlation coefficients (PCC) = 0.417) and DeepSEA (average PCC = 0.412)-based regulatory SNPs exhibited higher PCC values than the array SNPs (average PCC = 0.402) (Figure 6B and Appendix A). Notably, the DeepSATA-based regulatory SNPs displayed a higher PCC than the DeepSEA-based regulatory SNPs (Figure 6B). To ensure a fair comparison between the DeepSATA- and DeepSEA-based regulatory SNPs and the array SNPs, considering the difference in SNP numbers, we selected the top 32,451 SNPs from both the DeepSATA- and DeepSEA- based regulatory SNPs, matching the number of array SNPs. These SNPs were chosen based on having the largest absolute differences in reference and alternative allele values within their respective clusters. Among these subsets, as well as the array SNPs, a limited overlapping rate was observed. Specifically, 0.022% (20) SNPs were found to be present in all three sets, while 4.73% (4388) SNPs were common to both subsets of DeepSATA- and DeepSEA-based regulatory SNPs, as depicted in Figure 6C. We used the three clusters of SNPs in rrBLUP and found a consistent average PCC trend of 0.402 for the array SNPs, 0.405 for the subset of DeepSEA-based regulatory SNPs, and 0.417 for the subset of DeepSATA-based regulatory SNPs (Figure 6B and Appendix A).

## 3. Discussion

Although numerous DL tools have emerged to prioritize and annotate non-coding genomic variants, only a small fraction of them possess the ability for cross-species prediction. The limited availability and compromised nature of the epigenomic data for many non-model animals present significant challenges in investigating the genetic interpretation of these species. For the last decade, the compelling evidence indicates a high level of conservation in the tissue-specific regulation across species, encompassing TF proteins and their binding preferences [20,21,22]. Leveraging this wealth of conserved information, our introduced model, DeepSATA, not only achieves reliable cross-species prediction but also demonstrates superior same-species prediction performance. As a result, DeepSATA has the potential to make valuable contributions to the interpretation of the genetic codes underlying complex traits in diverse organisms having difficulties in generating epigenomics data.

We discovered that separating the training on 10 sets enabled us to concentrate on a subset of regions while still achieving meaningful outcomes (Appendix A). This observation is supported by the results demonstrating that the average AUC values across all samples of both mice and pigs were comparable between the DeepSATA model trained on 10% genomic regions (average AUC = 0.854 for mice, average AUC = 0.779 for pigs) and the DeepSATA model trained on 100% genomic regions (average AUC = 0.843 for mice, average AUC = 0.769 for pigs) (Figure 2A and Appendix A). A similar trend was also observed for the DeepSEA model (Figure 2B and Appendix A). Furthermore, similar to the performance superiority observed using 10% genomic regions (Figure 2), DeepSATA also exhibited superior performance compared to DeepSEA in terms of the average AUC values across the 10 groups of each sample in pigs and mice (Appendix A).

Despite being an effective approach, DeepSATA still has considerable potential for further improvement and enhancement. For instance, the TF binding sites utilized in our cross-species predictions were not always derived from experiments. The data in the JASPAR database [24] come from a combination of experimental and computational sources. The experimental data are obtained through reliable techniques such as chromatin immunoprecipitation and DNase I hypersensitivity, directly capturing the binding of TFs to DNA in real cells or tissues. In contrast, computationally predicted data are generated using algorithms that utilize known TF binding sequence patterns, such as position weight matrices, and statistical methods to scan the genome for potential binding sites. The precision of the predictions in JASPAR can fluctuate, introducing the possibility of both false positives and false negatives in the identification of TF binding sites. Furthermore, the predictive models utilized by JASPAR may not encompass the complete intricacies and subtleties of TF binding. Hence, in the future, as more experimental data become available, there is potential to enhance our model by incorporating it with these reliable datasets. Furthermore, to alleviate the computational burden, we only considered the top 10 primary TFs for subsequent training. However, it is important to note that this selection may not fully represent the actual scenario and could potentially overlook relevant TFs. Therefore, when the computational resources are sufficient, we would consider incorporating all primary TFs for training purposes. This would allow for a more comprehensive analysis, capturing the potential contributions of all relevant TFs in the model.

In the performance comparison between DeepSATA and DeepSEA based on different chromatin features from different species, we found that the relative improvement of the average AUC for mice (DeepSATA: 0.854; DeepSEA: 0.796) was the highest (7.21%). Considering that these two models were built based on deep learning algorithms, we hypothesized there may be relations between the amount of data and the final prediction accuracy. We calculated several metrics based on the underlying working principles of DeepSATA (Table 2). The first two metrics were the number of bins labeled as open chromatin regions (Bins in Table 2) and the average number of bases that overlapped with open chromatin regions in each bin (Average Coverage in Table 2). Increasing the former would expand the training data size while increasing the latter would enhance the consistency of the chromatin accessibility assignment with our binning strategy. Moreover, we conducted an in-depth analysis of a specific metric that directly addresses the TF binding configuration—an essential feature in DeepSATA. This metric measures the motif-based occurrence percentage of the top 10 TFs in relation to all TFs (top 10 motif contribution ratio (TF.Ratio) in Table 2). Since DeepSATA incorporates only the top 10 TFs during model training, a higher percentage indicates a greater ability to capture a substantial portion of the TF binding configurations. Consequently, this would contribute to improved prediction accuracy within DeepSATA.

As shown in the Table 2, the chicken species exhibited the poorest performance, potentially due to having the lowest number of bins. This limitation resulted in inadequate model training with a small amount of data. Conversely, among the remaining four species, which had a significantly larger number of bins, the mice showcased the highest average bin coverage rate and top 10 motif contribution ratio. These factors likely contributed to the mice achieving the best prediction performance outcome. As shown in Table 3, for different pig breeds, the average coverage rates of the Duroc and LW pigs were higher than for ES and MS, which may have contributed to a better alignment of the chromatin accessibility with the 200 bp binning strategy. Although the *Duroc pigs* had a lower top 10 motif contribution ratio and average coverage rate than the LW pigs, they outcompeted the LW pigs in the number of bins, which might explain the better prediction accuracy of this pig breed. The interactions among different factors in contributing to the prediction accuracy is complex. In this case, the higher number of bins might have compensated for the lower top 10 motif contribution ratio and average coverage rate. To further demonstrate the importance of the bin number in terms of the prediction accuracy, we further downsampled eight tissues of mice and humans and calculated the average AUCs. For mice, the average AUCs were 0.828 and 0.746 for DeepSATA and DeepSEA, respectively, lower than the respective values of 0.854 and 0.796 using all tissues (Appendix A). For human, the average AUCs were 0.739 and 0.729 for DeepSATA and DeepSEA, respectively, also lower than the respective values of 0.759 and 0.755 using all tissues (Appendix A).

Our cross-species predictions using the mice model have provided reliable results, showcasing the ability to transfer their regulatory grammars to other animals. From this, it can be speculated that our DeepSATA model places importance on both sequence features and regulatory patterns by incorporating additional TF preferences as high-dimensional information. In the initial training step, DeepSATA independently learned grammars from the direct sequence as well as from each TF. Using a max pooling strategy, it aggregated and summarized the significant grammars, resulting in a more comprehensive atlas of *cis*-regulatory grammars. Despite variations in the referenced sequences across different species, there is a possibility that the atlas of *cis*-regulatory grammars covering enriched motifs remains conserved. This presents an opportunity to correct mismatched sequence grammars by leveraging conserved enriched motifs in specific tissues. This observation is supported by our examinations, where several cross-species predictions exhibited superior performance compared to same-species predictions (Figure 3 and Appendix A). We hypothesize that this discrepancy may be attributed to the lower quality of the training data available for the target tissues in these specific cases. When the training data quality is compromised, DeepSATA may struggle to learn the conserved regulatory rules across different species. Conversely, high-quality data from other species may effectively capture these conserved rules. In such situations, the improved learning efficiency of the conserved regulatory rules compensates for the accuracy loss resulting from the inability of the cross-species prediction to capture species-specific regulatory rules. This hypothesis gains support from the observation that chicken tissues exhibit a higher frequency of cases where cross-species prediction outperforms same-species prediction, as compared to other animals. This trend is consistent with the findings presented in Table 2, which reveal that chickens had the lowest number of 200 bp open chromatin bins utilized for training DeepSATA. Additionally, we found that mice-based predictions of liver tissue in three other animals achieved the highest average AUC of 0.749 (Figure 3 and Appendix A). By extracting enhancer sequences covering the OCR of live tissue samples, we further investigated the conservation of the chromatin accessibility in the same tissues across different organisms. We observed that 49.9%, 48.3%, and 8.8% of the mice liver OCRs have conserved homologous sequences (liftover minMatch  =  0.8) with the genomes of pigs, cattle, and chickens, respectively (refer to Appendix A). Additionally, among these homologous sequences, 32.6%, 8.0%, and 7.0% show chromatin accessibility in pigs, cattle, and chickens, respectively (refer to Appendix A). Moreover, we observed that the functional conservation of conserved DNA sequences is significantly higher compared to that of random DNA sequences, with a *p*-value < 0.01 (refer to Appendix A). Finally, by mapping homologous sequences of each shared tissue from other animals to mice, we calculated the sequence similarity, defined as the ratio of homologous sequences with chromatin accessibility in each tissue of each species. We found that there was a strong correlation, with a PCC of 0.619 between the sequence similarity of open chromatin regions and the cross-species prediction performance (Appendix A). These encouraging results suggested that there existed conserved regulatory grammar for enhancers, which is likely to impact the cross-species prediction performance of DeepSATA, motivating us to further enhance DeepSATA as a more precise transfer learning model. In the future, our focus will be on exploiting the conserved *cis*-regulatory grammars while re-learning the biological context-specific *cis*-regulatory grammars, which can potentially provide insights into the manual design of regulatory element sequences.

However, there were also serval limitations regarding the DeepSATA model. Firstly, the whole genome was partitioned into 200 bp consecutive bins. Although this strategy would miss important sequences when they reside at the boundary of two 200 bp bins, it has been widely used, such as in DeepGRN [32] and DeepCAGE [33], and achieves overall effective and strong prediction performance, suggesting that this binning strategy could capture sufficient information to make reliable predictions. Regarding the training process of DeepSATA, the proper lengths of the consecutive bins and extractive sequences that cover the important sequences for a given dataset need to be investigated in future work. Secondly, the effectiveness of using DeepSATA to predict *cis*-regulatory mutations was illustrated only in pigs. As the publicly available datasets are limited, especially for those functional mutations or SNPs related to phenotypes, this restricts the utilization of DeepSATA in comprehensively unraveling the genetic mechanisms underlying complex traits. Nevertheless, we still have confidence that the livestock genetics community will gain significant benefits from its utilization, especially as more genome-wide association studies are conducted and their findings are published. Lastly, the interpretation of our DeepSATA needs to be enhanced. Although we tried our best to explain the reasonability of encoding TF configurations as higher dimensional space features, we only performed two further attempts. One was to emphasize the importance of extending the three-dimensional space. When we reduced the three-dimensional features into two-dimensional features by extending the original 4 features (represent one-hot encoding of A, T, C, and G) to 14 features, this implied that instead of incorporating TF configurations into higher-dimensional spaces beyond those encoded by DNA sequences, we had included them at the same level as the four potential DNA nucleotides. With this new strategy, and while keeping the same number of 14 features (4 basic nucleotides and 10 TFs), the prediction performance (average AUC) of DeepSATA decreased from 0.854 to 0.827 (Appendix A), suggesting that DeepSATA benefits from the original coding strategy that enables the extraction of more regulatory information. Another was to illustrate the effectiveness of utilizing the top 10 TF DNA binding motifs enriched in open chromatin regions, whereby we re-trained DeepSATA using the TF DNA binding motifs ranked 11–20. Under this scenario, the average AUC decreased from 0.854 to 0.847 (Appendix A), underscoring the importance of choosing the most relevant TFs for model training. DeepSATA is an ongoing development project that continues to evolve and improve. We are confident that with the aid of expanding publicly available datasets, we will be able to delve deeper into our epigenomic analysis. Our future endeavors include exploring the prediction of enhancer-promoter pairs by leveraging training data that comprise sequences and essential TF motifs of DNA elements pairs. These pairs will be accompanied by corresponding labels indicating the presence or absence of physical interactions between them. This approach will enable us to advance our understanding and capabilities in the field of epigenomic analyses.

Overall, we are confident that DeepSATA will serve as a valuable tool for annotating and prioritizing regulatory variants, thereby enhancing our understanding of their mechanisms in different tissues across diverse animal species. We believe that DeepSATA will make a significant contribution to deciphering economic traits in livestock and addressing complex diseases in humans.

## 4. Materials and Methods

### 4.1. Data Collection for Constructing the Model

We retrieved the ATAC-seq data for mice from the NCBI Sequence Read Archive (SRA) database using the accessions listed in Appendix A. The ATAC-seq data for humans were downloaded from the ENCODE project [34] using the accessions listed in Appendix A. Different breeds of pigs including Enshi (ES), large white (LW), and Meishan (MS) were considered in our study, and the ATAC-seq datasets for ES, MS, Duroc, and LW male pigs were derived from the Gene Expression Omnibus (GEO) database under accession number GSE143288 [35]. The male cattle and chicken ATAC-seq datasets were downloaded from https://farm.cse.ucdavis.edu/~ckern/Nature_Communications_2020/Peak_Calls/ (accessed on 11 December 2022) [36]. The Irreproducible Discovery Rate (IDR) [37] framework was used to handle ATAC-seq peak replicates with a threshold of 0.05.

We downloaded non-redundant transcription factor binding sites for vertebrata from JASPAR 2022 [24] in the “MEME” format for use as training data for pigs, cattle, and chickens. The binding sites for *Mus musculus* were collected for use as training data for mice. We detected motif patterns using FIMO (version 5.4.1) with the default *p*-value [25].

We obtained the UCSC-provided *full genome sequences* for *Mus musculus* (*mm10*), *Gallus gallus* (*galGal6*), *Bos taurus* (*bosTau9*), *Homo sapiens*
*(hg38*), and *Sus scrofa* (*susScr11*) from Bioconductor.

### 4.2. Model Training in DeepSATA

We used a Torch-based deep learning framework to implement DeepSATA. This model consisted of three convolution layers (320, 480, and 960 kernels), with each layer followed by the ‘ReLU’, ‘max-pooling’, and ‘dropout’ operations. The ‘sigmoid’ is applied to generate the outputs, which represent the probabilities of chromatin accessibilities. To prepare the input of training, we partitioned the whole genome into 200 bp consecutive bins. Bins that overlapped with ATAC-seq peaks by more than half of their length were labelled as positive 1, with negative labels of 0 otherwise. Considering the high burden of computing, we randomly shuffled these genomic bins using the ‘shuf’ command and selected 10% of them for training. For each training bin, a 1000 bp sequence centered on a 200 bp bin is extracted from a reference genome. Each bin is represented by a three-dimensional 1000 × 4 × (N + 1) matrix, with the first dimension representing the sequence centered around the 200 bp bin, the second dimension representing the four possible DNA nucleotides at each base (A, T, C, or T), and the third dimension represent the encoding schedules of the DNA sequence and the TF binding configurations. The third dimension consists of a series of successive two-dimensional layers; for example, layer 1 represents the one-hot encoding of the current DNA sequence, with vectors [1,0,0,0], [0,1,0,0], [0,0,1,0], and [0,0,0,1] representing the nucleotides A, T, C, and G, respectively. Each of the following layers (starting from layer 2 to layer *N* + 1) encodes the binding affinity and location of a specific TF. In layer *N* + 1, as depicted in Figure 1, TF *N* was predicted to bind with a consensus motif of ‘ATCGATCGATCG’. Given that the position probabilities for the A and C alleles at the first bp were 0.9 and 0.1, this site was encoded with [0.9,0,0.1,0]. Accordingly, the sub-matrix corresponding to the predicted binding site of TF *N* in layer *N* + 1 was populated by the position weight frequency of TF *N*, with vectors [0,0.8,0,0.2], [0.3,0,0.7,0], [0,0.4,0,0.6], [0.8,0,0.2,0], [0,0.7,0,0.3], [0.4,0,0.6,0], [0,0.1,0,0.9], [0.7,0,0.3,0], [0,0.6,0,0.4], [0.1,0,0.9,0], and [0,0.2,0,0.8] representing the other 11 sites, while the other entries were set to 0. To determine N in the training set, we collected non-redundant transcription factor binding sites from the JASPAR 2022 [24] database and recruited the FIMO (version 5.4.1) tool to identify motif patterns in sequences with the default *p*-value [25]. We ranked the TFs based on their corresponding motif presence over all input bins and selected the top *N* TFs. Due to the computational resource constraints, only the top 10 motifs were used to construct the model. During the optimization process, we set learning rates of 1 for mice, chickens, humans, and cattle and 0.5 for pigs. To maintain independence between the training, validation, and test sets, we used chromosome 1 as the validation set, chromosomes 2 and 3 as the test set, and the remaining chromosomes for training. The output of this model was a vector of probabilities representing the likelihood of a sequence comprising open chromatin regions in each bin. In the end, we assessed the performance of DeepSATA using receiver the ROC and AUC.

To reduce the computational burden during model training, we utilized a strategy of partitioning the whole genome into 10 non-overlapping sets. Subsequently, we randomly selected one set for model training. In the specific cases of pigs and mice, we conducted separate training on 10 sets for each species.

Out of the 10 training groups conducted on both pigs and mice, we identified the ones with the highest average AUC values across all samples. These selected models formed the basis for our subsequent analysis, which included cross-species predictions and SNP open-chromatin effect predictions. By employing this approach, we effectively optimized the training process while maintaining the predictive accuracy of our model.

### 4.3. Cross-Species Predictions

We employed the DeepSATA model, trained on mice, which demonstrated the highest average performance (AUC = 0.854) for predictions within the same species. We extended the application of this model to make cross-species predictions for pigs, chickens, and cattle. Specifically, we utilized the model trained on a particular tissue to predict the corresponding tissues in other animal species. The AUC serves as a metric to evaluate the performance of predictions.

### 4.4. Predicting Impacts of Variants on Open Chromatin

We collected large-scale whole-genome SNPs in over 2800 *Duroc pigs* from Yang et al.’s article [31]. These SNPs were sequenced by a Tn5-based method that has high accuracy and allows low sequencing coverage [31]. Next, we imputed missing SNPs with the Beagle (version 5.1) tool [38] and obtained over 11.6 million (11,633,164) SNPs. To determine the impact of an SNP on open chromatin, we tested genomic regions within 500 bp on two sides of this SNP and calculated the likelihood of the sequence with either the reference or alternative allele being linked to open chromatin regions separately. We then computed the probabilities of these two sequences. The threshold for distinguishing open chromatin from closed chromatin was determined based on the optimal balance between sensitivity and specificity measures. SNPs were considered regulatory when the alleles showed different open chromatin statuses.

### 4.5. Enrichment Analysis on SNPs That Are Significantly Impacting Open Chromatin

We obtained the GWAS data from pigQTLdb [39,40], specifically collecting the ‘association’ information. We used the ‘bedtools intersect’ tool (version 2.25.0) to identify the intersection of significantly impacting SNPs with genomic regions within a 2 Mb range centered at the midpoint of each GWAS signal [41]. Subsequently, we assigned these overlapping SNPs to the nearest transcription start sites of genes within a 1 Mb distance using the ‘bedtools closest-d’ tool (version 2.25.0) [41]. We utilized the hypergeometric test to calculate the *p*-values and enrichment scores for each trait (big category). All genes within 2 Mb genomic intervals were considered as the background genes.

### 4.6. Phenotype Predictions Using rrBLUP

We collected the genotype data for 2802 pigs from Yang et al.’s study [31]. After imputation using Beagle (version 5.1) [38], we obtained a total of over 11.6 million SNPs. Phenotype data for the back fat thicknesses of 1940 samples were also downloaded from Yang et al.’s study [31] via GigaDB [42]. To ensure coherence between the genotype and phenotype data, we employed BCFtools [43,44] to generate a subset of genotype data consisting of these 1940 samples. Next, we used GCTA (version 1.94.0) [45] software to evaluate the kinship among individuals with the parameter “--make-grm-alg 1”. Subsequently, we utilized the rrBLUP package [30] to predict the phenotype from the genotype with 5-fold cross validation. Finally, we used the PCC as the performance evaluation metric.

To assess the accuracy of the phenotype predictions using array data, we initially gathered genomic locations from GeneSeek porcine 50K SNP chips. As these locations were constructed using the *Sus Scrofa 10.2* assembly, we utilized the liftOver R package [46] to convert them to the *Sus Scrofa 11.3* assembly, which was consistent with the pig ATAC-seq data. We then isolated these positions from the entire genotype data using BCFtools [43,44] to create a subset containing only array loci. The remaining procedures are consistent with the above.

We collected the DeepSATA- and DeepSEA-based regulatory SNPs, as defined in Section 2.6. For their respective subsets, we calculated the absolute differences between the reference and alternative allele values. Next, we selected the top 32,451 SNPs from both DeepSATA- and DeepSEA-based regulatory SNPs, matching the number of array SNPs.

### 4.7. Sequence Similarity Calculation

We extracted genomic regions spanning 1000 bp on both sides of open chromatin bins in mice, pigs, cattle, and chickens. The number of these regions was calculated as the ‘total’. To convert these regions to the target species, we utilized the ‘liftOver’ tool with a parameter setting of the ‘minimum ratio of bases that must remap’ at 0.8. The resulting genomic regions were referred to as ‘mapped’. Using the ‘bedtools’ software (version 2.25.0), we compared the ‘mapped’ regions with the ‘total’ specific to a particular species. The number of overlapping regions were defined as ‘overlapped’. By dividing the ‘mapped’ regions by the ‘total’, we calculated the ‘mapped ratio’, and by dividing the ‘overlapped’ regions by the ‘mapped’ regions, we calculated the ‘overlapped ratio’. Additionally, the ‘overlapped ratio’ was also defined as the sequence similarity.

To evaluate the significance of these findings, we conducted 100 simulations. In each simulation round, we randomly selected the same ‘total’ number and length using the ‘bedtools shuffle’ (version 2.25.0). From these simulations, we obtained the numbers of ‘mapped’ and ‘overlapped’ regions and calculated the average ‘overlapped ratio’ and ‘mapped ratio’ across the 100 simulations, known as the average simulated ratios. To determine the significance, we compared the observed ratios to the simulated ratios. We calculated the fold changes by dividing the observed ratios by the average simulated ratios. Additionally, we calculated the *p*-value by determining the proportion of simulated ratios that were equal to or greater than the observed ratio, divided by 100.

## 5. Conclusions

This study has presented DeepSATA, a new deep learning-based sequence analyzer that incorporates the transcription factor binding affinity for cross-species prediction. Through an analysis of the open chromatin data of pigs, chickens, cattle, humans, and mice, we demonstrated the enhanced prediction accuracy of the chromatin accessibility and the reliable cross-species prediction capabilities of DeepSATA. Furthermore, we showcased its effectiveness in analyzing pig genetic variants associated with economic traits. Overall, DeepSATA represents a valuable tool that broadens our knowledge of the potential influence of non-coding variants on complex traits across different species.

## Figures and Tables

**Figure 1 ijms-24-12023-f001:**
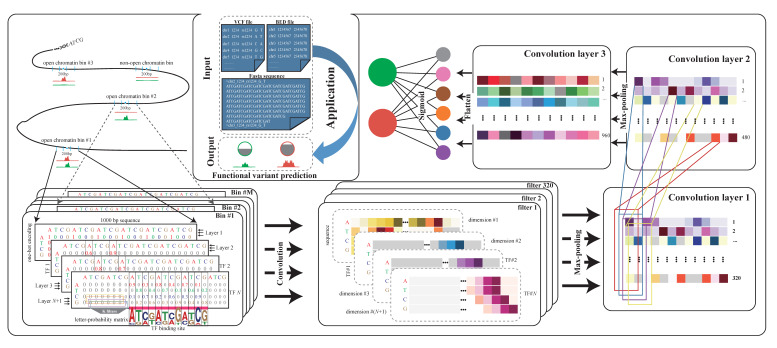
Schematic overview of the DeepSATA pipeline. Open chromatin bins (OCBs) are defined as being located within at least one ATAC peak region, and labeled as 1 if more than half of the 200 bp fall within the peak region and 0 if not. Each OCB is represented by a three-dimensional 1000 × 4 × (*N* + 1) matrix, which represents the sequence centered around the 200 bp bin, the four possible nucleotides, and TF binding configurations. For example, layer 1 represents the one-hot encoding of the current DNA sequence, with vectors [1,0,0,0], [0,1,0,0], [0,0,1,0], and [0,0,0,1] representing the nucleotides A, T, C, and G, respectively. Each of the following layers encodes the binding preference of a specific TF. In layer *N* + 1, the predicted binding location of TF *N* is marked by a prominent red line, accompanied by the position weight matrix logo of TF *N* placed beneath. The sub-matrix corresponding to the binding site of TF *N* is populated by the position weight frequency of TF *N*. Other entries of the layer are set to 0. Then, the three-dimensional input matrix is fed into a three-layer convolution neural network (CNN) with filters [320, 480, 960]. Each layer is followed by ‘ReLU’, ‘max-pooling’, and ‘dropout’ operations. Finally, ‘sigmoid’ is applied to generate the outputs, which represent the probabilities of being accessible at the chromatin level. For usage, DeepSATA supports input formats such as BED, VCF, and FASTA, and it generates functional variant predictions as outputs.

**Figure 2 ijms-24-12023-f002:**
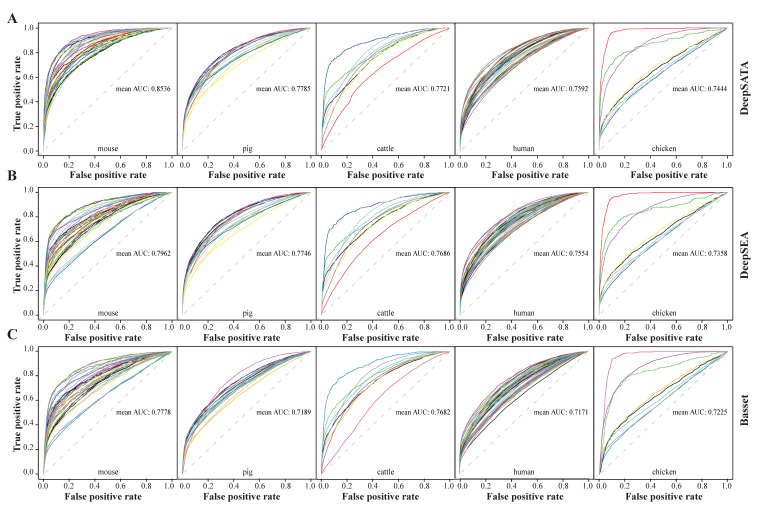
Comparison of ROC/AUC values among DeepSATA, DeepSEA and Basset: (**A**) AUC values of DeepSATA models trained on mice, pigs, cattle, humans, and chickens; (**B**) AUC values of DeepSEA models trained on mice, pigs, cattle, humans, and chickens; (**C**) AUC values of Basset model trained on mice, pigs, cattle, humans, and chickens.

**Figure 3 ijms-24-12023-f003:**
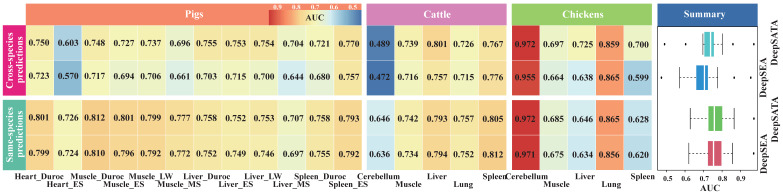
AUC values of cross-species predictions. AUC values of cross-species predictions for shared tissues of male mice, pigs, cattle, and chickens. ES denotes Enshi pigs. LW denotes *large white pigs*. MS denotes *Meishan pigs*.

**Figure 4 ijms-24-12023-f004:**
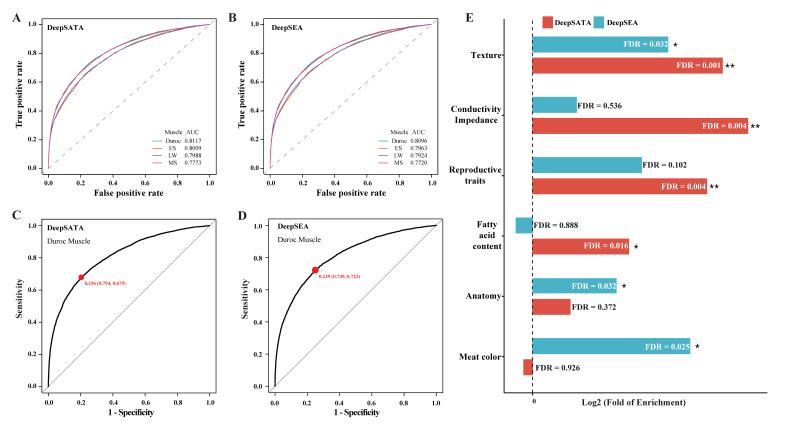
Identification of regulatory SNPs by DeepSEA and DeepSATA: (**A**) AUC values of DeepSATA models trained on different pig breeds; (**B**) AUC values of DeepSEA models trained on different pig breeds; (**C**) the optimal balance between sensitivity and specificity measures of the DeepSATA model trained on *Duroc pigs*; (**D**) the optimal balance between sensitivity and specificity measures of the DeepSEA model trained on *Duroc pigs*; (**E**) GWAS traits enriched in genes linked to DeepSATA-based and DeepSEA-based regulatory variants. Only traits with significant enrichment in at least one model are shown. Note: ** denotes FDR < 0.01 and * denotes 0.01 ≤ FDR < 0.05.

**Figure 5 ijms-24-12023-f005:**
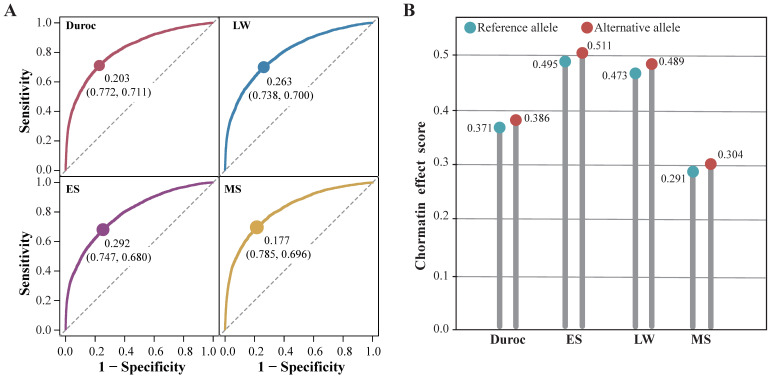
The chromatin effects of the selected SNP: (**A**) the optimal balance between sensitivity and specificity measures for the DeepSATA model trained on fat samples from Duroc, LW, ES, and MS pigs; (**B**) chromatin effect scores for reference (G) and alternative alleles (A) of an SNP positioned at chr5:66103958.

**Figure 6 ijms-24-12023-f006:**
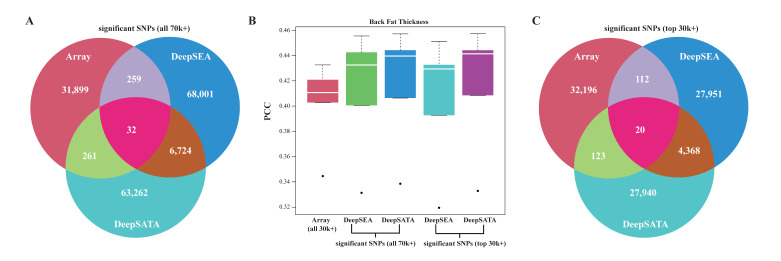
Performance of genomic prediction using rrBLUP with different sets of SNPs: (**A**) overlap among array SNPs and DeepSEA- and DeepSATA-based regulatory SNPs; (**B**) genomic prediction PCC values using array SNPs and DeepSEA- and DeepSATA-based regulatory SNPs, as well as the top 30k+ regulatory SNPs selected based on the greatest absolute differences between reference and alternative alleles, using the DeepSEA and DeepSATA models, respectively; (**C**) overlap among array SNPs and the top 30k+ SNPs from both the DeepSEA- and DeepSATA-based regulatory SNPs.

**Table 1 ijms-24-12023-t001:** Statistics of ATAC-seq data used in DeepSATA.

Species	Tissue Number	Breed Number	Gender Number	Sample Number	Mean OCR Number ^a^	JASPAR2022 Motif
*Mus musculus*	18	0	2	32	36,727	*Mus musculus*
*Sus scrofa*	5	4	0	16	124,190	Vertebrata
*Bos taurus*	8	0	0	8	84,463	Vertebrata
*Homo sapiens*	35	0	0	35	168,702	*Homo sapiens*
*Gallus gallus*	8	0	0	8	25,362	Vertebrata

^a^ Mean of the number of OCRs in each sample.

**Table 2 ijms-24-12023-t002:** Potential accuracy-affecting metrics and prediction accuracy of different models in different species.

	Bins ^a^	Average Coverage ^b^	TF.Ratio ^c^	DeepSATA	DeepSEA
mice	1,081,545	170.61	0.162	0.8536	0.7962
pigs	1,586,677	165.84	0.0946	0.7785	0.7746
cattle	1,872,585	167.14	0.0768	0.7721	0.7686
human	3,337,706	156.86	0.0763	0.7592	0.7554
chickens	570,905	166.76	0.0868	0.7444	0.7358

^a^ The number of bins labeled as open chromatin regions. ^b^ The average bin coverage represents the number of bases that overlapped with open chromatin regions in each bin. ^c^ Top 10 motif contribution ratio represents the percentage of the top 10 TFs in relation to all TFs.

**Table 3 ijms-24-12023-t003:** Potential accuracy-affecting metrics and prediction accuracy rates of different models in different pig breeds.

	Bins ^a^	Average Coverage ^b^	TF.Ratio ^c^	DeepSATA	DeepSEA
Duroc	884,195	167.08	0.01178	0.7891	0.7863
ES	1,307,825	163.45	0.0122	0.7725	0.7693
LW	748,865	167.86	0.0120	0.782	0.7766
MS	779,553	161.46	0.0120	0.7674	0.762

^a^ The number of bins labeled as open chromatin regions. ^b^ The average bin coverage represents the number of bases that overlapped with open chromatin regions in each bin. ^c^ Top 10 motif contribution ratio represents the percentage of the top 10 TFs in relation to all TFs.

## Data Availability

The DeepSATA model is freely available at https://github.com/mawenlong2016/DeepSATA (accessed on 10 June 2023).

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
