# Peer review of "DeepSATA: A Deep Learning-Based Sequence Analyzer Incorporating the Transcription Factor Binding Affinity to Dissect the Effects of Non-Coding Genetic Variants"

_ijms, 2023, doi:10.3390/ijms241512023_

Round 1

Reviewer 1 Report

In this manuscript by Ma et al, the authors introduced a tool name DeepSATA for analyzing and predicting cross-species sequence information based on another well-established work named DeepSEA. I think the authors have done a reasonable amount of work to show the efficiency and improvements of their work compared to the previous work in specific aspects. I have a few comments below:

1. The argument that regulatory rules are conserved across species serves as a strong assumption and is the fundaments of the ML models, which were trained on the data of a subset of species but can be used to predict other species, globally. Could the authors further elaborate more, in terms of more quantitative data/analysis, on the correctness of this assumption, in addition to merely the reference papers that they cited?

2. I acknowledge that given the nature of the deep-learning model, it's hard to do the model interpretation, but I still think it's necessary to show/validate why the authors' model outperforms DeepSEA. For example, the authors can potentially identify the key components/innovation of their model and do perturbations, and compare with the baseline to show which factors are really taking effect, instead of only presenting the final results without going deep into the underlying reason.

3. Enhancer-promoter pair prediction serves as an outstanding problem overall, I wonder how the authors' model behaves on that?

4. The authors tried their model based on ATAC-seq data, but potentially they could do more epi- data, like DNA methylation or epigenetics modifications etc. Have the authors considered those?

Reviewer 2 Report

In this manuscript, Ma W, et al., developed a novel deep learning-based sequence analyzer that incorporates transcription factor binding affinity for cross-species prediction of chromatin accessibility. The authors showed that DeepSATA, which they developed, improved the prediction accuracy of chromatin accessibility and achieve reliable cross-species predictions. The authors concluded that DeepSATA is a valuable tool to explore the epigenomic landscape of various species and pinpoint regulatory DNA variants associated with complex traits.

A well-considered and very interesting manuscript that is a development study of an analyzer that can be effective in predicting genomes across different species. However, there are some concerns in the manuscript. Therefore, I recommend the authors had better improve their manuscript according to the suggestions.

Major comments,

1.     The authors partitioned the whole genome into 200-bp consecutive bins and labeled 1 or 0 for each container by ATAC peak region. In this case, if a vital sequence resides at the boundary of two 200-base bins and is divided, could an important sequence be missed?

2.     In section 2.2, the authors explained that average AUC values of 0.854, 0.779, 0.772, and 0.744 for mice, pigs, cattle, and chickens, respectively in DeepSATA and of 0.796, 0.775, 0.769, and 0.736 ln DeepSEA. Why is the difference between the two analyzers greater in mice than in other species? If DeepSATA is so good at cross-species prediction, it is strange that the improvement is worse for other organisms than for mice, which have much data. Related to this, the authors described “both DeepSATA and DeepSEA models performed better for the Duroc pig breed compared to other pig breeds” (lanes 197-199). Could this be because there is more data on Duroc pigs than on other pigs?

3.     Are the results different between varieties or organizations because of the other numbers of data for each? If so, I think a table should be published showing the number of data for each breed and each tissue for each animal used in deep learning.

4.     In Section 2.5, the authors attribute the consistently higher scores for the A allele compared to the G allele across different pig breeds to the A allele promoting open chromatin. Please explain why you think it promotes open chromatin and not simply because of the genetically higher frequency of the A allyl in pigs.

Minor comments,

1.     I understand that Duroc is a breed of a pig; are MS and ES also?

2.     There are too many statements in the Discussion section regarding the description of the research methodology. There should be a sufficient explanation in the Methods and Results sections and omitted in the Discussion.

3.     In lanes 231 and 232, the authors described that the performance of cross-species predictions using the DeepSATA model was even better than that of same-species prediction. The reasons for this should be considered.

Reviewer 3 Report

DeepSATA may be useful for researchers who are interested in analyzing non-coding alterations in livestock. However, I have some concerns and they should be addressed.

Major comments

Figure 2, please include ROC/AUC for human dataset for more clarify. In addition, please include other models to compare for the model performance. I really do not understand why authors only evaluated the model performance against DeepSEA although it is a solid model. Based on my quick search, DeepTFactor (PNAS) and DeepGRN (BMC Bioinformatics) were found. This indicates that DeepSEA is not the only one to compare, other models as I abovementioned are also widely used in epigenetic analysis. 

    In addition, one good paper related to your project is reported in 2014 (Nat Genet. 2014, 46:1160-1165) and this paper was referred by the following paper (Nat Commun. 2019, 10:720). Furthermore, another paper also reported to identify non-coding rare variant association (Nat Commun. 2018, 9:3391). I believe comparing at least one more model is more informative rather than comparing certain tissues such as cerebellum, live, lung, spleen.

    For your information, Yong lab reported a benchmark of ROSE (and LILY) published in Cell (Cell. 2013, 153:307-19).

Figure 3 regarding cross-species prediction, it might be of great help if authors extract all enhancer region from mice, pig, cattle, and chicken and do a blast search to get an overall sequence similarity. Is the result of cross-species prediction associated with sequence similarity?

I have no idea why authors divided mouse data into male and female. What is the rational for it? Authors need to show the data from male/femal pigs, cattle, and chicken if authors would like to bring the sex dependent open chromatin information.

Minor comments

Due to the narrow research area, publicly available datasets are limited. Related to it, published data (mutations, SNP etc) liked to the phenotypes are also limited. Therefore, it could be of great help if authors mention their limitations in the discussion section.

Round 2

Reviewer 1 Report

The quality of the manuscript has been improved after the revision. Though there are more interesting outstanding problems, but I do acknowledge that those can be addressed in future endeavors. I can support the publication of the manuscript. 

Reviewer 3 Report

Although the authors did not provide any rebuttal/response letter for me, it looks like the revised manuscript clearly addressed my concerns. Therefore, no additional comments from me and I would like to congratulate on your efforts to this field.